# Identification of GB3 as a Novel Biomarker of Tumor-Derived Vasculature in Neuroblastoma Using a Stiffness-Based Model

**DOI:** 10.3390/cancers16051060

**Published:** 2024-03-05

**Authors:** Aranzazu Villasante, Josep Corominas, Clara Alcon, Andrea Garcia-Lizarribar, Jaume Mora, Monica Lopez-Fanarraga, Josep Samitier

**Affiliations:** 1Institute for Bioengineering of Catalonia (IBEC), The Barcelona Institute of Science and Technology (BIST), 08028 Barcelona, Spain; 2Department of Electronic and Biomedical Engineering, University of Barcelona, 08028 Barcelona, Spain; 3Biomedical Research Networking Center in Bioengineering, Biomaterials and Nanomedicine (CIBER-BBN), 28029 Madrid, Spain; 4Oncology Department, Pediatric Cancer Center Barcelona, Hospital Sant Joan de Deu, 08950 Barcelona, Spain; 5University of Cantabria-IDIVAL, 39011 Santander, Spain

**Keywords:** GB3, neuroblastoma, alternative vasculature, tumor-derived endothelial cells

## Abstract

**Simple Summary:**

Neuroblastoma (NB), a prevalent childhood cancer, presents challenges in treatment due to its cellular diversity and the presence of tumor-derived endothelial cells (TECs) associated with chemoresistance. We lack specific biomarkers for TECs, hindering effective therapies. We developed a stiffness-based in vitro platform simulating arterial and venous conditions to address this gap. Notably, adrenergic NB cells transdifferentiated into TECs where there was an arterial-like stiffness, while mesenchymal cells did not. This platform facilitated the identification of Globotriaosylceramide (GB3) as a novel TEC biomarker. Moreover, we harnessed Shiga toxin-functionalized nanoparticles for the specific targeting of GB3-positive cells, showing promise for future therapeutic strategies. Our study provides insights into NB heterogeneity, offers a predictive tool for assessing aggressiveness, and introduces potential targets for precision therapies.

**Abstract:**

Neuroblastoma (NB) is a childhood cancer in sympathetic nervous system cells. NB exhibits cellular heterogeneity, with adrenergic and mesenchymal states displaying distinct tumorigenic potentials. NB is highly vascularized, and blood vessels can form through various mechanisms, including endothelial transdifferentiation, leading to the development of tumor-derived endothelial cells (TECs) associated with chemoresistance. We lack specific biomarkers for TECs. Therefore, identifying new TEC biomarkers is vital for effective NB therapies. A stiffness-based platform simulating human arterial and venous stiffness was developed to study NB TECs in vitro. Adrenergic cells cultured on arterial-like stiffness transdifferentiated into TECs, while mesenchymal state cells did not. The TECs derived from adrenergic cells served as a model to explore new biomarkers, with a particular focus on GB3, a glycosphingolipid receptor implicated in angiogenesis, metastasis, and drug resistance. Notably, the TECs unequivocally expressed GB3, validating its novelty as a marker. To explore targeted therapeutic interventions, nanoparticles functionalized with the non-toxic subunit B of the Shiga toxin were generated, because they demonstrated a robust affinity for GB3-positive cells. Our results demonstrate the value of the stiffness-based platform as a predictive tool for assessing NB aggressiveness, the discovery of new biomarkers, and the evaluation of the effectiveness of targeted therapeutic strategies.

## 1. Introduction

Neuroblastoma (NB) is a malignant tumor derived from embryonic precursors of the sympathetic nervous system. Although NB is a rare cancer, it is the most common extracranial solid tumor in childhood. It frequently arises in the adrenal medulla, although the tumor can be found in any part of the anatomy within the sympathetic nervous system [1]. Metastatic presentation represents around 50% of all cases at diagnosis. The most common metastatic sites are the osteomedullary tissue (bone marrow/bone), the lymph nodes, the liver, the central nervous system, and the skin. Despite multimodality therapy, including surgery, chemotherapy, and immunotherapy, more than 50% of patients either progress during treatment or relapse after treatment [1].

Cellular heterogeneity is a hallmark of human neuroblastoma cell lines in vitro and in tumors from patients. More than four decades ago, NB cells in vitro were characterized as three distinct cellular phenotypic alternatives with different tumorigenic potentials: N-, S-, and I-type [2,3,4]. The N-type cell population (a neuroblastic phenotype) comprises small cells with scant cytoplasm and short neuritic processes that attach poorly to the substrate and grow as aggregates. N-type cells are highly malignant and efficiently form colonies in soft agar and tumors in mice [4]. The S-type (which is substrate-adherent) displays large, flattened cells that attach rapidly and firmly to the substrate and show contact inhibition. Interconversion between the N- and S-type phenotypes may occur in patients, as observed in cell cultures in seminal studies, but the mechanisms behind interconversion remain largely unknown [2,3,4,5]. Ultimately, the I-type displays a morphology that falls between that of the N- and S-types. It is proposed that the I-type cell serves as an intermediate stage in the conversion between the N- and S-types, or acts as a precursor to the N- or S-type cells with the ability for dual differentiation [2,3,4,5]. Studies have demonstrated that I-type cells exhibit the most tumorigenic phenotype in neuroblastoma, both in soft agar assays and in preclinical trials involving athymic mice [4]

Recent transcriptomic and epigenetic studies have shown that primary NB tumors comprise two types of tumor cells with divergent gene expression profiles [6,7]. Core regulatory circuitries, driven by specific transcription factors (TFs), that regulate the gene expression profiles in neuroblastoma have been identified for the traditionally recognized N- and S-type states. These states have now been redefined as the adrenergic and mesenchymal states, respectively, which can interconvert and resemble cells from different lineage differentiation stages [2,6,7]. Isogenic pairs of adrenergic and mesenchymal cells have revealed unique super-enhancer landscapes and super-enhancer-associated transcription factor (TF) networks specific to each cell type. These two distinct TF networks, likely responsible for orchestrating lineage control during normal development, exert a significant influence on the epigenetic regulation of neuroblastoma and contribute to the formation of intratumoral heterogeneity [6,7].

NB is also a highly vascularized tumor. Vascularization regulates growth, survival, and metastasis in multiple solid tumors. Tumor vessels can be formed through physiological-based mechanisms such as angiogenesis (sprouting and intussusceptive) and vasculogenesis [8,9]. Sprouting angiogenesis was the first tumor vascularization mechanism identified that generated new blood vessels from preexisting ones. Later, intussusceptive angiogenesis (or splitting angiogenesis) was observed in a human colon adenocarcinoma xenograft model, by which an existing vessel splits through the formation of intra-luminal tissue pillars [10]. Tumors can also generate vessels by an embryogenetic system known as vasculogenesis, which consists of the differentiation of vascular precursors into endothelial cells [8,9]. Tumors may also use cancer-specific strategies to obtain vascular networks such as vessel co-option that incorporates the existing vessels of the healthy organs into the tumor mass [8,9]. Furthermore, tumors have the capacity to employ alternative tactics, such as vascular mimicry and endothelial transdifferentiation, which operate independently of the host vascular endothelium. In the process of vascular mimicry, tumor cells organize themselves to create a vessel-like structure that lacks CD31 expression but shows positive periodic acid-Schiff (PAS) staining. Endothelial transdifferentiation involves cancer cells adopting endothelial cell characteristics, including the expression of CD31, and effectively transforming into tumor-derived endothelial cells (TECs) [8,9].

TECs are likely involved in chemoresistance in numerous tumors, including NB [11,12]. Current antiangiogenic strategies are designed to target the “classical mechanisms” but not the alternative vasculature, producing limited clinical efficacy and adverse outcomes [8,9]. This is partly due to the lack of targetable biomarkers for TECs besides the general endothelial-expressed proteins like CD31 or the stemness-related markers Oct-4 and TNC [13,14], which are also expressed in healthy cells. Thus, it is critical to identify new biomarkers for TECs to design more effective antiangiogenic therapies.

GB3/CD77 is a glycosphingolipid receptor whose synthesis involves sequentially adding specific sugar molecules to a lipid backbone facilitated by the enzyme lactosylceramide 4-alpha-galactosyltransferase encoded by the *A4GALT* gene. GB3 levels have been associated with invasiveness, metastasis, angiogenesis, and multidrug resistance. Studies have detected GB3 in endothelial cells within the tumor microenvironment of human cervical cancer, breast cancer, and head and neck cancer, or in murine neuroblastoma cell cultures, whereas blocking GB3 reduced tumor angiogenesis in mice models [15,16]. The expression of GB3 by TECs has not been reported.

A substantial technical limitation is the lack of predictive models, which has also hampered the identification of novel TEC markers. The presence of TECs in NB has been recapitulated in mouse models, which have constraints, including ethical issues (the “3Rs”) and cross-species differences [11], and human tissue-engineered models of a physiological size that also have limitations because of their dimensions, the amount of cell material, culture reagents, and ample space requirements [12].

Over the last decade, tissue-engineered models have successfully mimicked healthy human vasculature [17]. Different studies have demonstrated the role of the microenvironment in favoring vasculature formation in vitro [18,19,20], including substrate stiffness. Previous studies using polydimethylsiloxane (PDMS) substrates of varying degrees of stiffness demonstrated the influence of the extracellular matrix (ECM) rigidity on the arterial and venous differentiation of endothelial progenitor cells (EPCs) [20].

In this study, we aimed to recapitulate neuroblastoma TECs in vitro using a stiffness-based platform. Given the stemness features of neuroblastoma tumor cells, we hypothesized that NB cells may transdifferentiate on PDMS substrates as described for healthy EPCs. To this end, we designed a biocompatible PDMS-based platform mimicking human arterial and venous stiffness to induce endothelial transdifferentiation in vitro. Subpopulations of N-type and I-type cells (the adrenergic phenotypes) cultured in the arterial-like PDMS turned into TECs. In contrast, neither S-type cell lines (the mesenchymal phenotypes) studied transdifferentiated in any of the stiffness conditions tested. We investigated the possibility of identifying a new marker of TECs and focused on GB3. We confirmed that TECs expressed GB3. Utilizing the Shiga toxin’s inherent GB3-binding affinity, we prepared latex nanoparticles coated with its non-harmful subunit B, known as StxB, which demonstrated remarkable selectivity. Together, our results pave the way for targeting strategies for TECs to disrupt tumor growth and progression.

## 2. Materials and Methods

### 2.1. PDMS-Based Platform

Polydimethylsiloxane (PDMS) Sylgard 184 (Dow Corning) was blended in weight ratios of 1:75 and 1:70 (cross-linker:pre-polymer) to achieve equilibrium modulus values of Emod = 101.38 ± 6.22 kPa and Emod = 140.70 ± 5.24 kPa, respectively. For PDMS Sylgard 527, mixtures with proportions of 1:1 and 1:2 (A component:B component) were prepared, resulting in equilibrium modulus values of Emod = 3.39 ± 0.81 kPa and Emod = 7.09 ± 1.16 kPa, respectively.

#### Mechanical Characterization

To assess the mechanical properties, uniaxial compression tests were performed on the materials using a Zwick Z0.5 TN instrument (Zwick-Roell, Ulm, Germany) equipped with a 5 N load cell. Cylindrical samples, obtained with a 6 mm diameter biopsy punch, had their actual diameters and heights measured before testing. The experiments were conducted at room temperature, subjecting the samples to up to 30% of final strain (deformation) with a 0.1 mN preload force and a 20% min^−1^ strain rate. Stress–strain curves were derived from load–deformation measurements, and the compressive modulus values were calculated from the slope of the linear region corresponding to 10–20% of strain. Each material underwent a triplicate preparation, with three samples per preparation, and measurements were conducted in triplicate.

### 2.2. Chamber Slides Fabrication

Polydimethylsiloxane (PDMS) Sylgard 184 (Dow Corning, Midland, MI, USA) was mixed in a weight ratio of 1:10 (cross-linker:pre-polymer) and degassed into a desiccator for one hour. Then, the mixture was poured on a 150 mm culture dish and cured at 65 °C overnight. The PDMS was cut using a scalpel to obtain culture chambers of 20 mm × 60 mm (W × L) with three wells of 10 mm by 10 mm with a 0.80 mm thickness. Then, the PDMS chambers and glass slides (Thermo Scientific, Waltham, MA, USA; 26 mm × 76 mm (W × L), #1.5) were cleaned by sonication for 15 min in 70% ethanol and dried with a nitrogen gun. Clean glass slides were coated with Sylgard 184 with proportions 1:70 and 1:75 (cross-linker:pre-polymer) and Sylgard 527 with 1:1 and 1:2 (A component:B component). Finally, the PDMS chambers were bonded to coated glass slides using a plasma cleaner (Harrick, Fayette County, Georgia, PCD-002-CE) for 30 s at HP and 30 mm O_2_ flow.

### 2.3. Cell Culture 

Neuroblastoma cell lines. The SK-N-BE (2) cell line (sourced from the American Type Culture Collection, ATCC) and the SK-N-LP, LA1-5s, and SK-N-AS cell lines, generously provided by Jaume Mora’s Laboratory at HSJD, were cultured in an RPMI medium supplemented with 10% (*v*/*v*) of fetal bovine serum (FBS) and 1% of penicillin/streptomycin at 37 °C with 5% of CO_2_ in a humidified incubator.

*Live–Dead assay.* The samples underwent a 30 min incubation in an RPMI medium enriched with 2 μM of Calcein and 4 μM of ethidium homodimer-1 at 37 °C under 5% of CO_2_, adhering to the manufacturer’s instructions outlined in the LIVE/DEAD^®^ Viability/Cytotoxicity Kit (Molecular Probes, Eugene, OR, USA). The imaging of the samples was conducted using a fluorescence microscope (Olympus IX81 light microscope, Center Valley, PA, USA).

*MTS Cell Proliferation Assay.* The MTS cell proliferation assay in a monolayer was conducted utilizing the CellTiter 96^®^ AQueous One Solution kit (Promega, Madison, WI, USA, G3582) in accordance with the manufacturer’s instructions.

### 2.4. Flow Cytometry Analysis

To conduct a flow cytometry analysis, the cells grown in a six-well plate with PDMS coatings of various stiffness levels were trypsinized, gathered, and combined in a 15 mL Falcon tube, followed by centrifugation at 500× *g* for 5 min. After discarding the supernatant, the cells were stained with a primary antibody specific to CD31 (dilution 1:100, DAKO, Glostrup, Denmark, #M0823) in Hank’s Balanced Salt Solution (HBSS) supplemented with 2% of Fetal Bovine Serum (FBS) for 30 min on ice. After the primary antibody incubation, the cells underwent three washes with PBS, were centrifuged at 500× *g* for 5 min, and then exposed to secondary antibodies (goat anti-mouse Alexa Fluor 647 in a dilution of 1:500) in HBSS + 2% FBS on ice for 30 min. Following three PBS washes, the cells were resuspended in 250 µL of PBS and subjected to analysis using a flow cytometry Gallios instrument (Beckman Coulter, Nyon, Switzerland). Negative controls involved cells incubated without primary antibodies. Data analysis was carried out using Flowing Software (Turku Bioscience Centre, University of Turku, and Åbo Akademi University, Finland; https://bioscience.fi/services/cell-imaging/flowing-software/; accessed on 5 February 2024).

### 2.5. Immunofluorescence (IF)

For the immunofluorescence analysis, the cells were fixed using 10% of neutral buffered formalin (Sigma-Aldrich #HT5011, St. Louis, MO, USA) at room temperature for 15 min. After the fixation, the samples were permeabilized for 15 min with PBS containing 0.2% of Triton X-100 and subsequently blocked for 1 h with 5% of BSA in PBST (PBS + 0.1% Tween 20). Following this, the samples were subjected to an overnight incubation at 4 °C in a humid chamber with a primary antibody specific to CD31 (dilution 1:50, DAKO, #M0823, or 1:50, Abcam #ab28364), diluted in antibody diluent (Dako, S3022). GB3/CD77 (1:50, Biolegend, San Diego, CA, USA, clone 5B5) was used for co-staining. On the following day, the samples underwent three washes (5 min each) with PBST and were then incubated at room temperature in a humid, dark chamber for one hour with the appropriate secondary antibody (Thermo Fisher, Waltham, MA, USA; Goat anti-mouse-Alexa Fluor 647 or goat anti-rabbit-Alexa Fluor 488) and diluted 1:500 in the antibody diluent. Following the secondary antibody incubation, the samples were treated with Hoechst (dilution 1:5000) for 15 min at room temperature and visualized either through a fluorescence microscopy (Olympus IX81 light microscope) or a confocal microscopy (Zeiss LSM 800, Oberkochen, Germany).

### 2.6. Quantitative Real-Time PCR (qRT-PCR)

The isolation of the total RNA from the cells was accomplished using Trizol (Life Technologies, Carlsbad, CA, USA), following the guidelines provided by the manufacturer. Subsequently, the obtained RNA samples underwent treatment with “Ready-to-go you-prime first-strand beads” (GE Healthcare, Chicago, IL, USA) to generate complementary DNA (cDNA). A quantitative real-time PCR was conducted using the PowerUp SYBR Green Master Mix (Applied Biosystems #A25742). The mRNA expression levels were quantified using the ΔCt method, where ΔCt is calculated as (Ct of the gene of interest—Ct of Actin). The primers for the PECAM and *A4GALT* were sourced from the PrimerBank database (http://pga.mgh.harvard.edu/primerbank/; accessed on 5 February 2024).

### 2.7. Datasets for Genomics Analysis

The R2 Genomics Analysis and Visualization Platform (http://r2.amc.nl; accessed on 5 February 2024) was utilized to investigate the *A4GALT* mRNA levels in the neuroblastoma tumors, following previously described methods [21]. This online genomics analysis tool allows researchers to analyze publicly available microarray datasets. The MegaSampler analysis was performed using multiple datasets profiled on Affymetrix U133 (u133p2) arrays and normalized using MAS5.0. Two datasets, the cell lines and neuroblastoma tumors with or without *MYCN* amplified, were profiled to show the expression levels of the *A4GALT* (Gene ID: 53947). The data for the neuroblastoma cell lines originate from Versteeg’s laboratory and can be accessed through the GEO ID: GSE28019. The NB primary tumor data originates from Versteeg’s Laboratory study of 88 human NB samples and is available under the GEO ID: GSE16476.

### 2.8. Nanoparticles Functionalization and Neuroblastoma Cell Treatment

There were 400 nm of latex nanoparticles labeled in red (Abcam # ab269893), which were coated with the Shiga toxin non-toxic subunit B-STxB (Merck Life Science # SML0562, Darmstadt, Germany) following the manufacturer’s instructions. Briefly, the STxB stock was diluted to 0.1 mg/mL with the Reaction Buffer A and mixed with the latex nanoparticles for 15 min. After stopping the reaction, 1 μg of functionalized nanoparticles was added to the cells in a well of the chamber slide cultured for one week and incubated overnight. Then, the immunofluorescence was performed as indicated above using the GB3/CD77 (1:50, Biolegend, clone 5B5) as the primary antibody and goat anti-mouse-Alexa Fluor 488 (1:500, Thermo Fisher) as the secondary antibody.

## 3. Results

Following the same rationale to differentiate healthy EPCs into vascular vessels [20], we cultured NB cells on venous-like and arterial-like PDMS substrates to induce transdifferentiation into TECs. We prepared two different venous-like PDMSs using Sylgard 527 with proportions of 1:1 and 1:2 (A component:B component), corresponding to equilibrium modulus values of E_mod_ = 3.39 ± 0.81kPa (the lowest value reported for the venous tissue) and E_mod_ = 7.09 ± 1.16 kPa (the value published to induce venous differentiation in vitro), respectively (Figure 1A). Next, we fabricated the arterial-mimicking physiological stiffness substrates. We used Sylgard 184 with a composition of 1:70 (curing agent:base) and an E_mod_ = 140.70 ± 5.24 kPa for obtaining the highest described stiffness value for the mammal arteries. Sylgard 184 with a composition of 1:75 (curing agent:base) was used to obtain the optimum in vitro value of the substrate stiffness to differentiate the EPCs into arteries with an E_mod_ = 101.38 ± 6.22 kPa (Figure 1A). N-type cell cultures on top of the materials grew similarly for the four conditions in the RPMI medium but slower than in a plastic culture plate (with an elastic modulus of about 1 × 10^7^ kPa) (Figure 1B) and remained alive at least until day 7 as observed by the live–dead staining (Figure 1C).

Compliant substrates fail to induce expression of the PECAM/CD31 transdifferentiation marker in N-type cells, as depicted in Figure 2A,B. This lack of expression persists even on highly soft materials with an Emod = 0.2 kPa (Figure 2A). However, when cultured for 7 days on arterial-like PDMS substrates, the NB cells exhibit notable PECAM mRNA expression levels, surpassing those observed in plastic and soft controls (Figure 2A). The flow cytometry analysis identifies a distinct subpopulation of CD31+ cells originating from the 100 kPa and 140 kPa plates (Figure 2B).

For a detailed analysis of the transdifferentiated cells, immunofluorescence targeting CD31 was conducted. A specially designed device featuring chambers on a glass slide, coated with 1:70 and 1:75 Sylgard 184, was employed for this purpose (Figure 2C). As expected, the N-type cells failed to express CD31 when cultured on the glass control. However, the presence of tumor-derived endothelial cells (TECs) was confirmed in the 100 kPa and 140 kPa chamber slides (Figure 2D).

The venous PDMS platform also induced transdifferentiation (Figure 2B), but the subpopulation of TECs was smaller than the one obtained with the arterial-like PDMS substrates. Interestingly, the 7 kPa plates favored transdifferentiation better than the 3 kPa, as described for healthy EPCs differentiation.

Then, we asked whether the endothelial cell growing medium (EGM) would better induce transdifferentiation in the 100 kPa and 140 kPa substrates. We cultured the cells with an RPMI and EGM mix. Unfortunately, we detected a high level of cell death in the 100 kPa condition (Appendix A), and the values of the PECAM gene expression for both stiffnesses were comparable to those observed for the cells cultured with RPMI only (Appendix A).

To delve further into the capability of NB cells to transdifferentiate on the PDMS platform, we investigated the possibility of obtaining TECs from the other two NB cell line types. An I-type (an adrenergic phenotype) cell line was cultured on the PDMS substrates for 3 and 7 days. In contrast to the N-type cells, the I-type proliferated similarly in the plastic plates and the venous-like PDMS but faster than in the arterial-like PDMS material (Appendix A). All conditions were nontoxic as tested by live–dead assays on days 3 and 7 (Appendix A).

An analysis through a flow cytometry of the cell subpopulations expressing CD31 revealed a notable induction of endothelial transdifferentiation in the I-type cell phenotype on both the 100 kPa and 140 kPa substrates (Figure 3A,B). Significantly, the 100 kPa substrate proved to be the most effective in prompting this transition. As expected, the immunofluorescent images corroborated the presence of CD31 expression on the cell membrane within the subpopulations of cells cultured on the arterial-like PDMS material (Figure 3C).

Finally, we examined the S-type NB cell line LA1-5s (mesenchymal phenotype), which also proliferated slower in the arterial-like PDMS substrates (Appendix A) and did not show cell death in any of the five materials tested (Appendix A). Unlike the N-type and I-type cells analyzed, the LA1-5s cell line did not express PECAM/CD31 in any of the stiffness conditions tested (Figure 4A,B).

We analyzed a second S-type cell line to explore whether the inability to transdifferentiate the LA1-5s depended on the cell line characteristics. The SK-N-AS cells were cultured on the PDMS platform, and proliferation and survival were checked (Appendix A).

Interestingly, the SK-N-AS cells proliferated similarly in all conditions and slower than in plastic plates (Appendix A). As reported for the LA1-5s cell line, SK-N-AS showed PECAM/CD31 expression comparable to the negative control of cells cultured on plastic (Figure 4C,D).

Once the model to transdifferentiate NB cells in vitro was set up and validated, we focused on the potential expression of GB3 by TECs. Specifically, we chose the N-type model for these studies because the subpopulation of TECs obtained was the highest among all the cell lines tested.

GB3 is a globoside, so gene expression analysis is unsuitable for detecting GB3 cellular levels. However, determining mRNA levels of *A4GALT*, the enzyme that catalyzes the transfer of galactose to lactosylceramide to form globotriaosylceramide, can provide realistic information about the biosynthesis of GB3.

Comparing cells and tumors from the patients, we observed that *A4GALT* expression levels were higher in NB tumors with *MYCN* amplification in the monolayer (Figure 5A). The N-type cells SK-N-BE(2) are *MYCN* amplified and expressed significantly higher levels of *A4GALT* when cultured on the arterial-like stiffness substrates compared to the conventional plastic culture plates, strengthening the capability of the model to mimic some of the NB patient’s features (Figure 5B). Interestingly, *A4GALT* levels were higher in the 100 kPa plates, which was the optimal condition for obtaining TECs in vitro. Therefore, we used the 100 kPa condition as the model system to evaluate the potential co-expression of CD31 and GB3. The immunofluorescence of CD31 and GB3 confirmed that the TECs co-expressed both markers (Figure 5C), which points to GB3 as a novel and selective biomarker for TECs and underscores its potential as a promising therapeutic target.

We employed a novel approach to investigate further and utilize the unique GB3 expression as a targeting mechanism. The Shiga toxin is well-documented for its strong binding affinity to GB3 receptors [22]. By exploiting the natural affinity of the Shiga toxin for these receptors, researchers have developed toxin-based nanocarriers to target various cancer types [22,23,24,25] As a proof of concept, we decided to functionalize latex nanoparticles of 400 nm with the non-toxic subunit B (StxB) of the Shiga toxin for specific targeting of GB3-expressing cells. Notably, the confocal studies demonstrated a high degree of selectivity, with these nanoparticles exhibiting a strong affinity for GB3-positive cells while showing minimal interaction with GB3-negative cells (Figure 6).

## 4. Discussion

Since the TECs are not present in traditional plastic supports, we aimed to design and build a model to reproduce the endothelial transdifferentiation in vitro and to identify novel targets expressed in TECs. To develop the simplest model, we proposed to tune the stiffness of the substrate, focusing on the reported stiffness of very well-studied vasculatures like arteries (50–150 kPa) and venous tissues (3–50 kPa) [26]. We also used the established stiffness to differentiate endothelial progenitors in vitro (7 kPa for venous tissues and 128 kPa for arteries) [20].

We demonstrated that the adrenergic cell phenotypes (the N-type and I-type) transdifferentiated in the stiffer substrates in the range of arteries. Interestingly, the reported stiffness for the collagenous bone (a common site for NB metastasis) is about 100 kPa [27], similar to the value observed for the best-evaluated condition for the transdifferentiation of NB cell lines. Our data suggest that only adrenergic phenotype NB cells hold the capacity to transdifferentiate.

Low tumor vascularity in NB is associated with a favorable prognosis [28]. Vascularization enables tumors to obtain oxygen and nutrients, favoring growth and promoting dissemination to distant organs [8,9]. Thus, the transdifferentiation into TECs could favor malignant progression linked to tumor aggressiveness and metastatic potential. We only found a subpopulation of TECs in those NB cell phenotypes defined as tumorigenic, while the mesenchymal phenotype (the S-type) lacked transdifferentiation capacity.

Significantly, conventional cancer models lack the cell heterogeneity observed in clinical samples. When cells adapt to a plastic plate, crucial subpopulations involved in surviving the challenging conditions of the human body, such as those orchestrating tumor vasculature formation or maintaining the cancer stem cell niche, are eliminated [12,29]. While these subpopulations constitute only a small percentage within the tumor cell mass and its microenvironment, their significance for tumor development cannot be overstated. Consequently, it can be argued that drugs impacting cancer cells in plastic culture models may not effectively target the essential subpopulations found in native tumors, as these are inadequately represented in the in vitro setting. Therefore, it becomes critical to replicate in vitro those cell populations playing a significant role in tumor maintenance in vivo to assess the efficacy of specific treatments against them.

Therefore, we can propose the stiffness-based PDMS platform as a simple method to recapitulate cell heterogeneity and predict NB aggressiveness and metastatic potential based on the capacity to transdifferentiate into TECs. In addition, the platform can be used as a previous drug-testing step before more complex models to discard ineffective treatments in a fast, cheap, and high-throughput way compatible with standard pharmaceutical industry equipment and protocols.

Recent research has highlighted the role of GB3 in cancer. Studies have shown that GB3 expression is upregulated in certain cancer types, including neuroblastoma, although its role in neuroblastoma has not been as extensively studied as in other cancer types. Using the platform, we identified GB3 as a novel marker present in TECs. The overexpression of GB3 was not only in the TECs but also in the bulk of the tumor, which makes GB3 a perfect candidate for targeted therapies and diagnostic purposes. Researchers have explored several strategies to target GB3 as a potential therapeutic approach against cancer [30]. The most recent strategy consists of GB3 targeting using the Shiga toxin, a bacterial toxin that binds specifically to GB3 receptors, particularly in combination with nanoparticles [22]. As a proof of concept and for translational purposes, we prepared latex nanoparticles coated with the non-toxic subunit B and demonstrated the possibility of GB3-positive cell targeting. Although further research is necessary, this strategy could be especially beneficial for pediatric patients to minimize off-target toxicity by selectively targeting TECs and NB cells with GB3 expression. However, GB3 and its association with cancer is a relatively emergent area of investigation, and clinical applications are not yet widespread.

## 5. Conclusions

In conclusion, our study aimed to design and develop an in vitro model that mimics endothelial transdifferentiation to identify novel target molecules expressed in tumor-derived endothelial cells. By tuning the substrate stiffness to mimic the arterial and venous tissues and the endothelial progenitor differentiation conditions, we observed that the most aggressive neuroblastoma cell phenotypes (the N-type and I-type) predominantly transdifferentiated in stiffer substrates, which resembled the arterial stiffness. This finding suggests that the NB cells’ aggressiveness may be related to their capacity to transdifferentiate into TECs, indicating a potential link between tumor aggressiveness and vascularization. Notably, the stiffness-based PDMS platform allowed for recapitulating the cell heterogeneity observed in clinical samples, often lacking in traditional cancer models. Moreover, our study identified GB3 as a novel marker in TECs, making it a promising candidate for targeted therapies and diagnostic purposes. While further research is required to fully understand the role of GB3 in neuroblastoma and its clinical applications, our findings support the utility of the stiffness-based platform as a tool for predicting NB aggressiveness and evaluating targeted treatment approaches.

## Figures and Tables

**Figure 1 cancers-16-01060-f001:**
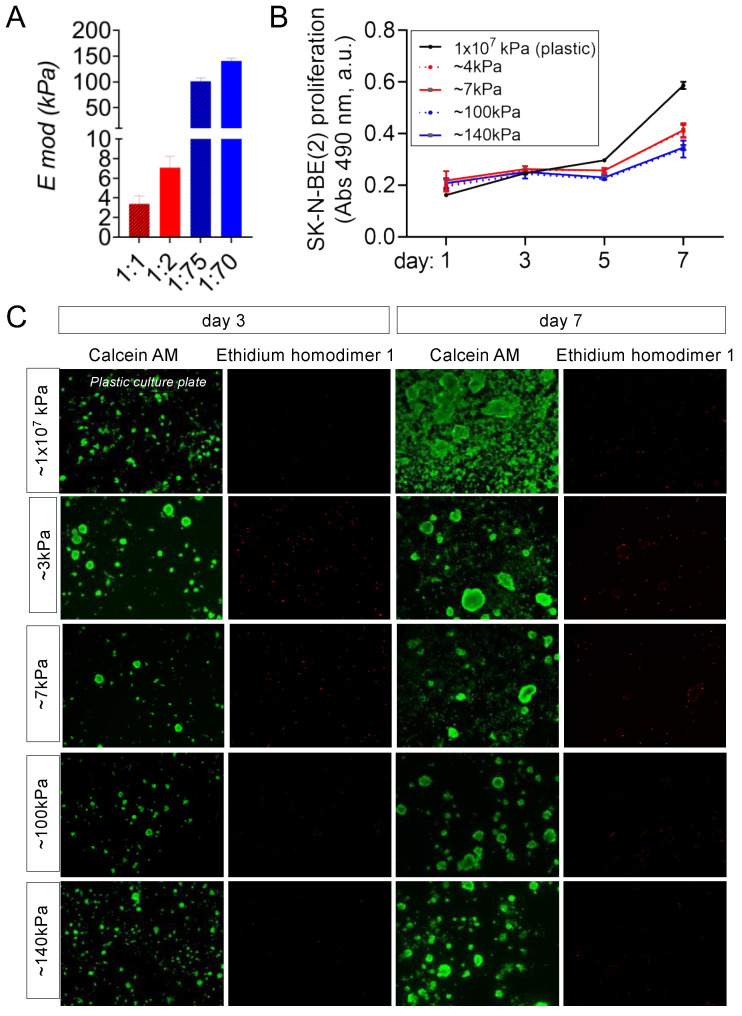
Characterization of the venous-like and arterial-like PDMS materials. (**A**) Equilibrium modulus (E mod) of the venous-like PDMS substrates (red) prepared with Sylgard 527 (proportions of 1:1 and 1:2) and arterial-like PDMS (blue) made of Sylgard 184 with a composition of 1:75 and 1:70. (**B**) Proliferation of N-type neuroblastoma cells (SK-N-BE(2)) cultured on venous-like PDMS (red) and arterial-like PDMS substrates (blue), compared to plastic dishes for the indicated time points (*n* = 6). A colorimetric method was used (MTS assay). Absorbance (Abs) at 490nm (arbitrary units, a.u.) is proportional to cell number. (**C**) Live–dead fluorescence images of SK-N-BE (2) cells cultured on the indicated substrates for 3 and 7 days *(n* = 3): calcein staining (green—live cells) and ethidium homodimer-1 staining (red—dead cells).

**Figure 2 cancers-16-01060-f002:**
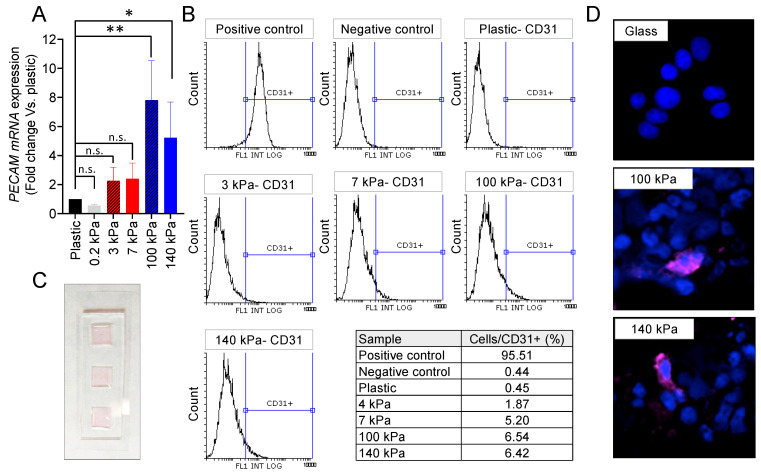
Endothelial cell transdifferentiation of N-type neuroblastoma on venous-like and arterial-like PDMS materials. (**A**) mRNA expression levels of PECAM in SK-N-BE (2) cells cultured for 7 days on substrates with varying stiffness. The fold change was calculated by normalizing to the GAPDH levels in the individual samples and then to the corresponding levels in the cells cultured in plastic dishes. Data are presented as the average ± SD (*n* = 3). Statistical significance was determined using the two-tailed Student’s *t*-test: * *p* < 0.05; ** *p* < 0.01; ns, not significant. (**B**) FACS analysis of the CD31 endothelial marker in SK-N-BE (2) cells cultured for 7 days on the specified PDMS substrates (*n* = 4). HUVEC served as the positive control, while cells incubated without primary antibodies were used as negative controls. (**C**) Design of the cell culture chamber slide, constructed with PDMS Sylgard 184 in a weight ratio of 1:10 (cross-linker:pre-polymer). The chamber dimensions are 20 mm × 60 mm (W × L) with the cell culture wells measuring 10 mm by 10 mm and a thickness of 0.80 mm. (**D**) Representative images illustrating CD31 expression (magenta) in SK-N-BE (2) cells cultured for 7 days on the designated substrates, with nuclei stained using Hoechst 33342 (blue) (*n* = 4). The culture took place in the chamber slides.

**Figure 3 cancers-16-01060-f003:**
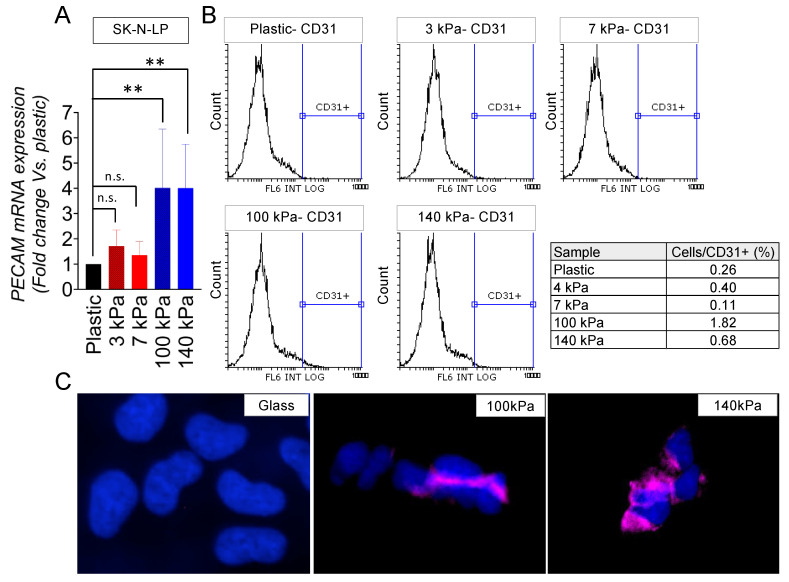
Endothelial transdifferentiation of I-type neuroblastoma on venous-like and arterial-like PDMS materials. (**A**) Quantification of PECAM mRNA levels in SK-N-LP cells cultured on substrates with varying stiffness for 7 days. The fold change was computed by normalizing to the GAPDH levels in the individual samples and then to the corresponding levels in the cells cultured in plastic dishes. Results are presented as the average ± SD (*n* = 3). Statistical significance was determined through the two-tailed Student’s *t*-test: ** *p* < 0.01; ns, not significant. (**B**) Flow cytometry analysis of the CD31 endothelial marker in SK-N-LP cells cultured for 7 days on the specified PDMS substrates (*n* = 4). (**C**) Illustrative images displaying CD31 expression (magenta) in SK-N-LP cells cultured for 7 days on the designated substrates, with nuclei counterstained using Hoechst 33342 (blue) (*n* = 4). The cell culture was conducted within the chamber slides.

**Figure 4 cancers-16-01060-f004:**
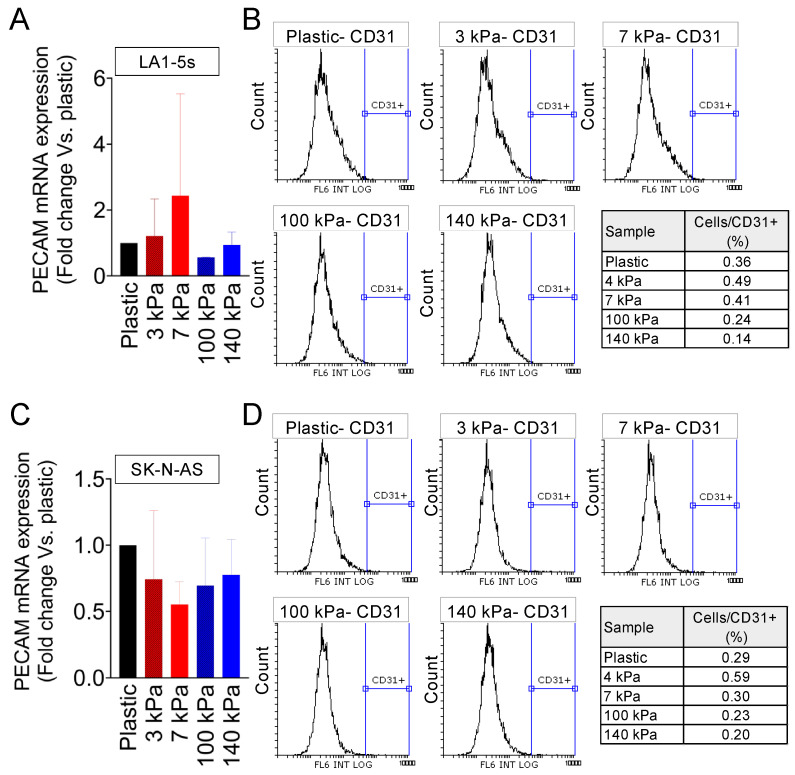
Endothelial transdifferentiation of S-type neuroblastoma on venous-like and arterial-like PDMS materials. (**A**) Quantification of PECAM mRNA levels in LA1-5s cells after 7 days of culture on substrates with different stiffness. The fold change was calculated by normalizing to the GAPDH levels in the individual samples and then to the corresponding levels in cells cultured in plastic dishes. Results are presented as the average ± SD (*n* = 9). (**B**) Flow cytometry analysis of the CD31 endothelial marker in LA1-5s cells following a 7-day culture on the specified PDMS substrates (*n* = 6). HUVEC served as the positive control, and cells incubated without primary antibodies were used as negative controls. (**C**) PECAM mRNA levels in SK-N-AS cells cultured for 7 days on substrates with varying stiffness. The fold change was determined by normalizing to the GAPDH levels in the individual samples and then to the corresponding levels in cells cultured in plastic dishes. Data are presented as the average ± SD (*n* = 3). (**D**) Flow cytometry analysis of the CD31 endothelial marker in SK-N-AS cells after a 7-day culture on the specified PDMS substrates (*n* = 3).

**Figure 5 cancers-16-01060-f005:**
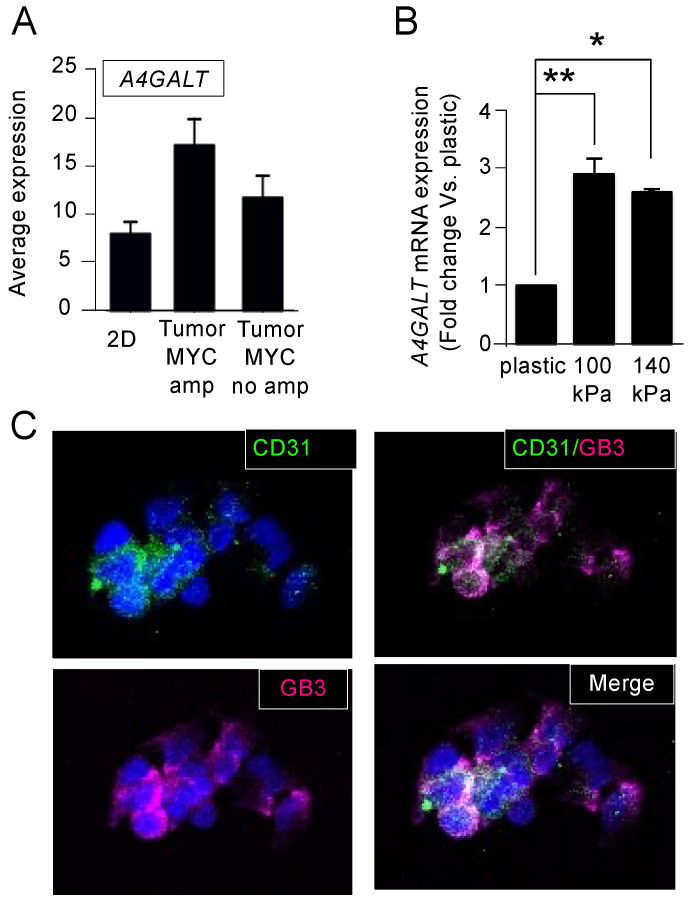
Expression of Gb3 in tumor-derived endothelial cells. (**A**) Comparison of *A4GALT* mRNA average expression of human neuroblastoma cells in monolayer (2D) and human neuroblastoma tumors with or without amplified *MYCN* using BoxPlot analysis based on publicly available expression array data. The analysis was conducted with the AMC Onco-genomics software tool. (**B**) mRNA levels of *A4GALT* in SK-N-BE (2) cells cultured on the indicated substrates for 7 days. The fold change was calculated by normalizing the GAPDH levels within each sample and then relative to the *A4GALT* levels in the cells cultured on plastic culture dishes. Data are presented as the average ± standard deviation (SD) from three independent experiments (*n* = 3). Statistical significance was determined by the two-tailed Student’s *t*-test: * *p* < 0.05; ** *p* < 0.01. (**C**) Representative images illustrating the presence of CD31 (green) and Gb3 (magenta) in SK-N-BE (2) cells cultured on chamber slides coated with the 100 kPa substrate for 7 days; the nuclei are counterstained with Hoechst 33342 (blue) (*n* = 4).

**Figure 6 cancers-16-01060-f006:**
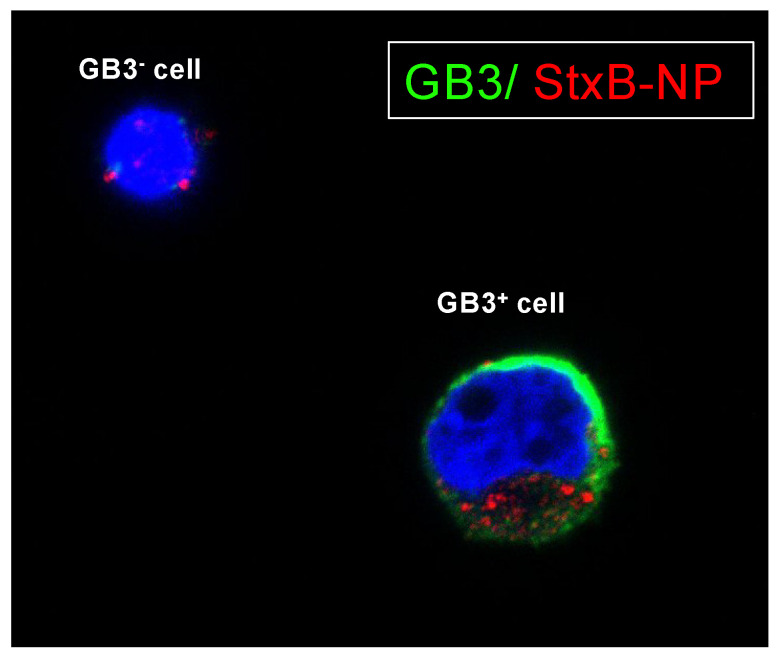
Targeting GB3-expressing cells with StxB-functionalized latex nanoparticles. Representative confocal images of SK-N-BE(2) cells cultivated on chamber slides coated with a 100 kPa substrate for 7 days. Cells were subjected to overnight incubation with StxB-functionalized latex nanoparticles. GB3 expression is visualized in green; latex nanoparticles are labeled in red; nuclei are counterstained with Hoechst 33342 (in blue) (*n* = 3).

## Data Availability

The datasets from and/or analyzed during this current study are available from the corresponding author upon reasonable request.

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
