# Peer review of "Identification of GB3 as a Novel Biomarker of Tumor-Derived Vasculature in Neuroblastoma Using a Stiffness-Based Model"

_cancers, 2024, doi:10.3390/cancers16051060_

Round 1
Reviewer 1 Report
Comments and Suggestions for Authors
Authors reported the detailed method for detecting tumor derived endothelial cells using mouse neuroblastoma cell line, while paying attention to the BG3. Then they showed that identifying GB3 in tumor derived endothelial cells could be the new marker for predicting neuroblastoma. Study is well established. And this marker may become the new target for treating aggressive neuroblastoma, so it is interesting, and I think this manuscript is acceptable for the journal. However, I found two abbreviations which did not provide full spelling. Authors should add the spelling which showed in below;
Please add the full spelling of "GB3" (Page 1 L20) and ECM (P3 L 125).
Author Response
Please, see the attachment

Reviewer 2 Report
Comments and Suggestions for Authors
Dear Authors,
I would like to commend the authors on their manuscript titled "Identification of GB3 as a novel biomarker of tumor-derived vasculature in neuroblastoma using a stiffness-based model." The study is intriguing and well-designed. However, there are a few points that require clarification.
Could the authors please provide information on the positive control cells for CD31 and GB3? As you are aware, confirmation of immunofluorescence staining with positive controls is crucial to ensure that the observed staining is not a false positive.
Thank you for your attention to this matter.
Best regards,
Author Response
Please, see the attachment
